# Adhesion of Oral Bacteria to Commercial d-PTFE Membranes: Polymer Microstructure Makes a Difference

**DOI:** 10.3390/ijms23062983

**Published:** 2022-03-10

**Authors:** Gabrijela Begić, Mirna Petković Didović, Sanja Lučić Blagojević, Ivana Jelovica Badovinac, Jure Žigon, Marko Perčić, Olga Cvijanović Peloza, Ivana Gobin

**Affiliations:** 1Department of Microbiology and Parasitology, Faculty of Medicine, University of Rijeka, 51000 Rijeka, Croatia; gabrijela.begic@uniri.hr (G.B.); ivana.gobin@uniri.hr (I.G.); 2Department of Medical Chemistry, Biochemistry and Clinical Chemistry, Faculty of Medicine, University of Rijeka, 51000 Rijeka, Croatia; mirnapd@medri.uniri.hr; 3Department of Surface Engineering of Polymer Materials, Faculty of Chemical Engineering and Technology, University of Zagreb, 10000 Zagreb, Croatia; slucic@fkit.hr; 4Department of Physics, University of Rijeka, 51000 Rijeka, Croatia; ijelov@phy.uniri.hr; 5Centre for Micro- and Nanosciences and Technologies & Center for Artificial Intelligence and Cyber Security, University of Rijeka, 51000 Rijeka, Croatia; marko.percic@riteh.uniri.hr; 6Department of Wood Science and Technology, Biotechnical Faculty, University of Ljubljana, 1000 Ljubljana, Slovenia; jure.zigon@bf.uni-lj.s; 7Department of Anatomy, Faculty of Medicine, University of Rijeka, 51000 Rijeka, Croatia

**Keywords:** bacterial adhesion, oral bacteria, polytetrafluoroethylene (PTFE), d-PTFE membrane, polymer microstructure, guided bone regeneration

## Abstract

Bacterial contamination of the membranes used during guided bone regeneration directly influences the outcome of this procedure. In this study, we analyzed the early stages of bacterial adhesion on two commercial dense polytetrafluoroethylene (d-PTFE) membranes in order to identify microstructural features that led to different adhesion strengths. The microstructure was investigated by X-ray diffraction (XRD), differential scanning calorimetry (DSC), and Fourier transform infrared (FTIR). The surface properties were analyzed by atomic force microscopy (AFM), scanning electron microscopy (SEM), and surface free energy (SFE) measurements. Bacterial properties were determined using the microbial adhesion to solvents (MATS) assay, and bacterial surface free energy (SFE) was measured spectrophotometrically. The adhesion of four species of oral bacteria (*Streptococcus mutans*, *Streptococcus oralis*, *Aggregatibacter actinomycetemcomitas*, and *Veilonella parvula*) was studied on surfaces with or without the artificial saliva coating. The results indicated that the degree of crystallinity (78.6% vs. 34.2%, with average crystallite size 50.54 nm vs. 32.86 nm) is the principal feature promoting the adhesion strength, through lower nanoscale roughness and possibly higher surface stiffness. The spherical crystallites (“warts”), observed on the surface of the highly crystalline sample, were also identified as a contributor. All bacterial species adhered better to a highly crystalline membrane (around 1 log_10_CFU/mL difference), both with and without artificial saliva coating. Our results show that the changes in polymer microstructure result in different antimicrobial properties even for chemically identical PTFE membranes.

## 1. Introduction

Bacterial adhesion to biomaterial surfaces is an immensely complex phenomenon of great importance from the theoretical and the practical point of view [1,2]. Oral bacteria have the ability to adhere to all surfaces within the oral cavity, both biotic and abiotic. The adhesion is a crucial step in biofilm formation, which in turn causes the development of pathological processes, clinical infections, and implant failure [3,4,5,6,7]. It was reported that 60–70% of all healthcare-associated infections are attributed to biofilm infections in implantable medical devices [5].

In a simplified view, the bacterial adhesion to any surface proceeds in four distinct stages: (1) transport to the surface; (2) initial (weak, reversible) adhesion; (3) attachment and (4) biofilm formation [2,8]. The stages are driven by a variety of non-specific and specific interactions, along with an array of biological processes. Stage one, the transport to the surface, is determined by bacteria Brownian motion, sedimentation, liquid flow or active bacterial movement. This planktonic state transfers to the surface-associated state when bacteria encounter and sense the surface by mechanosensing [9]. Stage two, the initial adhesion, is dominated by omnipresent non-specific interactions, i.e., short- and long-range physicochemical forces, the combination of which results in either attraction or repulsion of bacteria from the surface. On the other hand, the specific interactions are species-specific: they entail the role of bacteria surface appendages (fimbriae, flagella), and specific ligand-receptor interactions between the bacteria and substratum [2]. Specific extracellular proteinaceous components called adhesins act as ligands, while complementary receptors on substratum can be proteins, glycoproteins, polysaccharides, etc. Specific interactions are dominant in stage 3; they enable the transition of a weak, reversible adhesion to a permanent attachment.

A unified theory that would explain all aspects of bacterial adhesion still does not exist. Non-specific interactions can be viewed through the thermodynamic approach, classical DLVO (Derjaguin, Landau, Verwey, and Overbeek) theory, or extended DLVO theory, the latter being the most accepted [2,10,11,12]. In this theory, the bacterial adhesion is described as a balance between attractive long-range Lifshitz–van der Waals forces (i.e., London, dipole-dipole and dipole-induced dipole forces), repulsive or attractive electrostatic forces, and short-range Lewis acid–base (AB) interactions. AB interactions take into account the possibility of some surface moieties to act as electron acceptors/donors (i.e., Lewis acid/base, respectively) and represent the overwhelming contribution compared to Lifshitz–van der Waals and electrostatic forces. On the other hand, Lifshitz–van der Waals forces have a large 50 nm range, while AB interactions extend to not more than 5 nm. This approach is reflected in the AB model of surface free energy (SFE) calculations, where a total SFE is regarded as a sum of the Lifshitz–van der Waals, the acidic and the basic component, as opposed to solely non-polar and polar component in other models. The AB contribution to the total SFE is given as a square root of their product; hence, if only one component is present, it will interact with a complementary component of another phase but will not contribute to the total SFE. The predictions of this theory are confirmed and used by the MATH (Microbial Adhesion To Solvents) assay, in which diethyl ether and chloroform are used as solvents with exclusively acidic and basic SFE components, respectively, in order to measure bacteria SFE acid/base characteristics [11]. The SFE of the bacteria and the substratum, which are correlated to their wettability and hydrophobicity, are known to be an important influence on bacterial adhesion [5,13]. Indeed, Zhang et al. [14] found an inverse proportionality between the difference of bacterial and substratum SFE and the adhesion strength, for smooth model surfaces. Smoothness, i.e., the roughness of substratum surface, is also known as one of the major factors in bacterial adhesion, but opposing reports render it impossible to draw any general conclusions [5,7,15]. Surface roughness affects bacterial adhesion interdependently with surface topography and surface stiffness. Finally, fluid dynamics is a commonly neglected but—due to its effect on bacterial mechanosensing—important factor, especially for adhesion in the oral cavity [5].

After tooth extraction, the resorption of alveolar bone needs to be minimized to obtain enough volume for the implantation of a new tooth. One of the strategies to rebuild the lost bone is guided bone regeneration (GBR), where the use of a membrane is an essential component of the treatment [16]. The membranes are needed to prevent the penetration of the surrounding gingival epithelium and oral bacteria into the post-extraction cavity and to avoid interference by non-osteogenic cells [17]. Non-resorbable membranes typically used in clinical practice are either expanded (e-) or dense (d-) polytetrafluoroethylene (PTFE) membranes that may or may not contain titanium [18,19]. d-PTFE membranes are designed to prevent bacterial colonization by their fibril-free structure and low porosity, and even though bacterial migration on the internal surface has been documented, an optimal percentage of a newly formed bone has been reported at the end of the GBR procedure [19]. Even though bacterial contamination is directly correlated with gingival recession and reduced gain of clinical attachment level [20,21], the topic of bacterial adhesion on commercial membranes is generally poorly investigated. For PTFE membranes in particular, the rare studies that do exist are focused on the comparison of e- and d-PTFE [15] or PTFE with other materials [22].

PTFE is one of the most inert polymers used in biological systems; it is stable in host tissues and does not provoke an immunological response [16]. In addition, in GBR its widespread use within the biomedical field includes stents, stent grafts, cosmetic implants, tissue scaffolding, and orthopaedic products, etc. [23,24,25]. Along with excellent thermal stability, chemical resistance, and insulating properties (but high wear rate [26]), it is renowned for its low friction coefficient and low SFE, i.e., hydrophobicity [27,28]. Those properties arise from its chemical composition and structure: PTFE is built from linear helical (–CF_2_–CF_2_–) chains, in which tightly packed fluorine atoms make them look like smooth cylindrical rods and act as a protective layer. Small changes in helical conformations lead to room-temperature solid–solid transitions: triclinic (phase II) to hexagonal (phase IV) around 19 °C and hexagonal to pseudo-hexagonal (phase I) around 30 °C. It is a highly crystalline polymer, with a melting point of around 330 °C. Various crystallite morphologies have been observed, depending on the fabrication process and thermal history [28,29,30]. Crystalline features and the degree of crystallinity greatly affect PTFE physicochemical and mechanical properties.

This study was inspired by our empirical observation that the bacterial growth differed on two kinds of commercial d-PTFE membranes used in alveolar ridge preservation procedures. Both kinds are pure dense PTFE, without reinforcements, containing surface macro-roughness features (ridges vs. dents), thus differing only in physical features and polymer microstructure. One of them (designated PTFE-W) is well-established in clinical use [15,19], while the other one (PTFE-B) is new but proved to be effective in alveolar ridge preservation [31]. The effect of diverse microstructural features of chemically identical materials on bacterial adhesion has been observed for other materials; for example, bacterial adhesion was affected by crystalline vs. amorphous phase on Ti-based material for orthopaedic implants [32].

Therefore, the aim of this study was to uncover the main contributions in the polymer microstructure responsible for the observed differences in bacterial adhesion on two kinds of commercial d-PTFE membranes. We analyzed the early stages of adhesion of four bacterial species, namely, *Streptococcus mutans*, *Streptococcus oralis*, *Aggregatibacter actinomycetemcomitans*, and *Veilonella parvula*, due to their being early colonizers, known for rapid adhesion to oral cavity surfaces and their ability to promote the adhesion of other bacterial species as well as biofilm production [22,33]. We compared adhesion strength on membranes with and without artificial saliva coating, mimicking acquired oral pellicle. The paper is structured as follows: first, we present the analysis of PTFE membranes microstructure and surface, then the analysis of bacteria surface properties, followed by the results of adhesion strength measurements on uncoated and coated membranes. The results enabled the elucidation of main membrane features that led to enhanced bacterial attachment and growth.

## 2. Results

### 2.1. PTFE Microstructure

#### 2.1.1. ATR-FTIR Spectroscopy

Attenuated total reflection (ATR)-Fourier transform infrared (ATR-FTIR) spectra were recorded to examine membranes’ microstructure and possible differences in the composition of samples. As shown in Figure 1a, both spectra are dominated by two bands characteristic for PTFE, located at 1209 cm^–1^ and 1151 cm^–1^. The bands are associated with C-F symmetric and asymmetric stretching, respectively, and are considered to be insensitive to crystallinity [34]. An array of bands ranging from 500–650 cm^–1^, attributed to CF_2_ wagging (503 cm^–1^), deformation (554 cm^–1^), and rocking (637 cm^–1^), are also present. The bands in the PTFE-W spectrum were sharper and better resolved. However, the main difference between the spectra was observed around 750 cm^–1^, where wide low-intensity bands are present only in the PTFE-B spectrum. Those bands are associated with the polymer amorphous phase [35,36]. This implies a lower degree of crystallinity in PTFE-B membranes compared to PTFE-W. Additionally, neither of the spectra contain any additional bands, suggesting there are no double bonds, no noticeable branching, no impurities, or any traces of degradation, scissions, or oxidation [25,37].

#### 2.1.2. X-ray Diffraction

As shown in Figure 1b,c, the main diffraction peak, originating from (100) crystal plane, was observed at the similar position for both samples, namely, at 18.13° and 18.05° for PTFE-W and PTFE-B, respectively, indicating hexagonal (phase IV) unit cell with similar lattice parameters (d-spacing was 0.4893 nm and 0.4915 nm, respectively). Two low-intensity peaks at around 31.6° and 36.7° are assigned to (110) and (107) crystal planes, respectively [38,39]. The most pronounced difference between the spectra is the markedly narrower and higher (100) peak of the PTFE-W membrane (intensity = 12,454 vs. 1004 for PTFE-B). In addition, the PTFE-B spectrum displays a pronounced background, which originates from the amorphous part of the polymer (Figure 1d). The results indicate a higher degree of crystallinity in the PTFE-W membrane compared to PTFE-B, which is in agreement with the FTIR results. The average crystallite size, as calculated from full-width half maximum of the main peak using the Scherrer formula, was larger for the crystallites in the PTFE-W membrane. The average crystallite size was 50.54 nm and 32.86 nm for PTFE-W and PTFE-B, respectively. Note that this method gives the “apparent” crystallite size, i.e., the crystallite-size distribution and crystallite shapes are not taken into account [40].

#### 2.1.3. Differential Scanning Calorimetry

Figure 2a shows the first heating step for PTFE membranes in a dry state (solid lines) and after soaking in artificial saliva for 48 h (dotted lines). The most pronounced feature is the difference in melting endotherms of the two samples, both in intensity and position. The area of the peak is proportional to a degree of crystallinity, which was determined as 78.6 ± 0.5% for PTFE-W and 34.2 ± 1.4% for PTFE-B. This result is in accordance with XRD and IR measurements. The melting points were 349 °C and 326 °C for PTFE-W and PTFE-B, respectively. The lower melting point of PTFE-B is attributed to a smaller average crystallite size, as determined with XRD measurements (chapter 3.1.2). Another difference is a more pronounced shoulder on the PTFE-W endotherm (insert in Figure 2a), indicating two distinct populations of crystallites. Two smaller low-temperature endotherms are attributed to solid-solid transition between PTFE crystalline phases: triclinic (phase II) to hexagonal (phase IV) around 19 °C and hexagonal to pseudo-hexagonal (phase I) around 30 °C [36,41,42]. In addition to higher intensity due to a larger degree of crystallinity in PTFE-W, the positions and shapes of those endotherms are similar for both membranes. Upon cooling (Figure 2b), the crystallization exotherm of similar shape and intensity can be observed around 300 °C for both membranes. The second heating step (Figure 2c) displayed pronounced differences compared to the first heating step for the PTFE-W membrane, and now the thermograms of the two membranes resembled each other. The previously higher melting point of PTFE-W crystallites lowered and coalesced with the PTFE-B melting point (which remained unchanged), the shoulder disappeared, and the melting enthalpy diminished. This implies that crystal domains disordered completely upon melting, and the new crystallites formed from a melt were smaller and more uniform in size compared to pristine PTFE-W crystallites. The degree of crystallinity also decreased. This phenomenon has already been observed by Ranjbarzadeh-Dibazar et al. [43]. The differences in PTFE-W and PTFE-B microstructures disappeared after the melting, which emphasized how different they de facto were in the as-received state.

It is known that small molecules can diffuse between the PTFE crystallites, which leads to the weakening of the van der Waals forces within the crystallites, thus changing their properties [43]. For that reason, we wanted to examine whether the membranes’ properties change when used as intended in the oral cavity. Differential scanning calorimetry (DSC) thermograms of the membranes previously soaked in artificial saliva for 48 h are shown in Figure 2 with dotted lines. The results demonstrated that crystallites were not affected during this time. Both low-temperature transition endotherms and the melting endotherms remained similar in shape, intensity, and position for both membranes.

### 2.2. Surface Roughness and Topography

#### 2.2.1. Microscopic Surface Roughness by Atomic Force Microscopy

Atomic force microscopy was employed to examine microscale surface roughness on both sides of the PTFE membranes. The results showed that (a) the roughness of PTFE-B is an order of magnitude larger than the roughness of PTFE-W and (b) the roughness is side-dependent for both PTFE-W and PTFE-B, although less so for PTFE-B (Figure 3 and Table 1). The surface on both sides of PTFE-W was homogenous, with peak-to-valley heights around 300 nm on side 1 and around 200 nm on side 2 (Figure 3a,b, respectively). On the other hand, the microscale images revealed that two distinct types of areas can be distinguished on both sides of the PTFE-B, which was confirmed with a digital microscope (Figure 3c). The first type of area contains macro-waviness with peak-to-valley heights of approximately 4 μm and peak-to-peak distance of around 10 μm. The second type is smoother (peak-to-valley heights around 1 μm) with no apparent waviness. Note that even the smoother PTFE-B areas have larger RMS values than PTFE-W. PTFE-based materials and coatings containing different levels of macro-, micro, and nano-scale roughness are not uncommon since this provides one way of tailoring materials’ end properties [28,44].

#### 2.2.2. SEM Analysis

Scanning electron microscopy (SEM) micrographs provided a closer look at surface topography and revealed new topographical features. Two distinct areas, with or without macro-waviness, revealed by AFM on the PTFE-B surface are also visible on SEM micrographs (Figure 4a,b, respectively). An additional feature not recorded by AFM was the presence of fine fibers distributed over both types of areas but, interestingly, only on a single PTFE-B side. Figure 4c–e shows different magnifications of the other, fiber-free side. Larger magnifications are taken in order to examine the nano-scale features of the surface. They revealed that the nano-scale topography of PTFE-B consisted of a dense assembly of cigar-shaped PTFE crystallites, the so-called dendrites [28,29]. On the other hand, the micrographs of PTFE-W surfaces showed the existence of larger spherical crystallites, distributed heterogeneously over both sides of the PTFE-W membrane (Figure 4f–h). It has been shown that such beadlike structures occur when the concentration of the polymer solution, from which the material is prepared, is sufficiently low [30]. This type of crystallites, so-called “warts”, has been observed in several cases, for instance when they were used as a means of PTFE texturing [28,45]. Larger magnifications (Figure 4i,j) revealed the coexistence of “warts” (yellow arrows) and dendrites (red arrow), which has been reported before [45]. The dendrites are similar in shape but larger in size compared to the ones seen on the PTFE-B surface. These findings explain larger average crystallite size determined by XRD and dual melting peak with a high-temperature shift observed on DCS thermograms for PTFE-W membrane. The crystallization during the subsequent cooling step obviously failed to recreate the “warts”, as demonstrated by similar second DSC heating curves for both membranes.

### 2.3. Contact Angles and Surface Free Energy Parameters

Surface free energy is a net force exerted upon the molecules on the surface of the material as they are surrounded by air on one side and the bulk material on the other. The total SFE is a sum of several components, in general dispersive (i.e., non-polar) and polar, the determination of which proved to be important for understanding multiple phenomena that include wettability and adhesion [46]. For solids, due to the low mobility of solid molecules, it is “a notoriously difficult task” [1]. It is determined indirectly, usually through measurements of contact angles of three different liquids that are then used in various mathematical approaches based on the famous Young equation [46,47,48,49,50]. It is well known that contact angle is affected by surface roughness, so the measured contact angle is referred to as “apparent” (*θ*’) as opposed to “ideal” (*θ*). The apparent and ideal contact angles are related by the Wenzel equation cos*θ*’ = *r* cos*θ*, where *r* reflects the surface roughness as the ratio of surface area and geometric area [51,52,53]. For surfaces with heterogeneous surface roughness, *θ*’ and *θ* are related as given in the Cassie equation [28,52]. Currently, those relations are considered oversimplified, since it was found that contact angles on rough and heterogeneous surfaces depend solely on solid-liquid interactions at the three-phase (air-liquid-solid) contact line, and not those within the contact perimeter [50]. The value of the parameter *r* proved to be difficult to obtain [51,53]; fortunately, a general observation (in accordance with the Wenzel equation)—stating that contact angles > 90° increase by roughening the surface—is well-established. It has been shown that even nano-scale roughness severely affects the contact angles on PTFE membranes [28,30,44,52,54,55]. As mentioned above, the apparent contact angles are used in various models, the most common being Owens–Wendt’s method [56], Wu’s method [57], and the acid–base approach (van Oss–Chaudhury–Good method) [10]. The former two yield the dispersive and the polar SFE components, while the latter determines the equivalent Lifshitz–van der Waals (dispersive) and Lewis acid/Lewis base (polar) components. Many validity issues have been raised due to the accumulation of results showing that the obtained SFE values severely depend on the choice of the probe liquids and due to other fundamental shortcomings [1,46,47,50]. Nonetheless, this is still a broadly used method for solid surfaces SFE determination, and thus we will regard the obtained membranes SFEs as relevant but orientational values. On the other hand, bacterial SFE was determined spectrophotometrically, as proposed by Zhang et al. [14], in a study that demonstrated that the difference between spectrophotometrically determined bacterial SFE and classically determined (i.e., by contact angle measurements) substratum SFE is inversely proportional to adhesion strength.

Static contact angle values of water, formamide and diiodomethane, the three probe liquids for SFE calculations, are given in Table 2. Static water contact angles (WCA) are also used as a measure of surface hydrophobicity, although advancing and receding angles should be included for a full description of surface hydrophobicity [54,58]. WCA for both PTFE-W and PTFE-B were larger than the theoretical value for a perfect PTFE surface (106.94°) [59], but values ranging from 108° [60] up to “superhydrophobic” angles (static *θ* > 150°) can be found in the literature [58,61]. Such dissipation of the WCAs reflects its dependency on surface topography, which extends even to nano-level features [52,62]. Our results showed greater differences between the two sides of a particular membrane than between the membranes themselves. We ascribe this to micrometre-scale topological entities visible on SEM micrographs. It is plausible that “warts” and threads present on one side of PTFE-W and PTFE-B, respectively, could cause the creation of air pockets in the PTFE-water interface. This greatly affects the WCA on all surfaces and is known as the Cassie–Baxter state [5,61,63]. This explanation is supported by the well-known phenomenon that for hydrophobic surfaces the water does not penetrate into “valleys” of topological entities with dimensions of less than several micrometres [62]. With formamide and diiodomethane, the PTFE wettability expectedly increases (i.e., *θ* diminishes), since the dispersive components of the liquids increase from water to formamide to diiodomethane (Table 2).

Total surface free energy values of PTFE membranes, obtained using the Owens–Wendt (O–W), Wu and acid–base (A–B) models, are shown in Figure 5. The value of the polar component obtained by the O–W and Wu models was zero (Appendix A), which is in accordance with the common notion that the SFE of an ideal PTFE membrane results solely from dispersive component [47,64]. Note that this implies that PTFE surfaces cannot exhibit any dipolar interactions. The dispersive components obtained by the A–B model (i.e., Lifshitz–van der Waals component, *γ^LW^*) were close to those obtained by the O–W model, but the A–B model also revealed a weak basic (electron-donor) component for one side of each membrane (Appendix A). This implies that one side of each membrane can contribute the acid–base type of interaction to interfacial interactions. The values of the dispersive component are in good agreement with values reported by Zielecka et al. [65] but lower compared to other reports [56,66,67], which is a consequence of the larger contact angles, as explained above.

However, our SFE measurements were performed primarily to establish the difference in SFE between PTFE membranes and a particular kind of bacteria, since it was shown that bacterial adhesion on smooth surfaces is (roughly) inversely proportional to SFE difference: the greater the SFE difference between the bacteria and surface, the more adhesion can be expected [14]. For that reason, bacterial SFE is also presented in Figure 5. It is apparent that the differences between values of membranes’ SFE obtained by different models are an order of magnitude lower compared to differences between membranes’ and bacterial SFE. The SFEs obtained for *S. mutans*, *S. oralis* and *V. parvula* were similar, while *A. actinomycetemcomitans* deviated with a markedly larger value. In other words, the difference between bacterial and membranes SFE was the largest for *A. actinomycetemcomitans*, while lower and similar for the rest of bacterial species. 

### 2.4. Microbial Adhesion

#### 2.4.1. Microbial Adhesion to Solvents (MATS)

The MATS assay is a popular non-contact-angle-based method to measure bacterial hydrophobic and Lewis base/acid properties, using hexane and chloroform, i.e., diethyl ether, respectively [11]. For the bacteria used in this study, the results showed that while all species exhibited similar hydrophobicities, their acid/base properties differed (Figure 6A). S. mutans had the lowest adhesion both in chloroform and diethyl ether, demonstrating its poor ability to act either as electron donor (Lewis base) or electron acceptor (Lewis acid). For *S. oralis*, *A. actinomycetemcomitans* and *V. parvula*, the adhesion to diethyl ether was inversely proportional to adhesion to chloroform; the adhesion to chloroform increased in a given direction. This indicated that *V. parvula* had the largest elector-donor capabilities (the strongest basic character), while *S. oralis* had the largest elector-acceptor capabilities (the strongest acidic character).

Unlike polar interactions, the Lewis acid–base approach of viewing surface-bacteria interactions is complementary, i.e., the acid–base character of a bacteria is able to contribute to interfacial interactions only if the complementary character is present on the surface [1]. Note that the perfect PTFE surface does not possess any acid–base character, but our measurement showed weak basic (electron-donor) character for one side of each membrane (Appendix A).

#### 2.4.2. Microbial Adhesion to Membranes

In order to examine the early phase of bacterial adhesion, as a crucial first step in biofilm formation, bacterial colonization was measured after 4 h on uncoated PTFE-W and PTFE-B surface and surfaces coated with artificial saliva. The results are shown in Figure 6B. The following observations can be drawn: (a) for both uncoated and coated surfaces, all the bacteria adhered better to PTFE-W; (b) *A. actinomycetemcomitans* displayed the weakest adhesion among bacterial species on all surfaces; (c) the strongest adhesion was established for *S. oralis*, followed by *V. parvula*, *S. mutans* and *A. actinomycetemcomitans*.

The results also showed that the coating of the membranes with artificial saliva affected the strength of early adhesion of all bacterial species, but with different magnitude and signs depending on the species. Nonetheless, the overall ranking of species remained unchanged. The adhesion strength was lowered the most on both membranes for *A. actinomycetemcomitans*, followed by *S. oralis* (also weakened adhesion on both membranes). For *V. parvula*, the adhesion was reduced on PTFE-W while slightly increased on PTFE-B. Only *S. mutans* adhered stronger on coated membranes.

Note that this technique measures the overall adhesion, thus the differentiation of adhesion for a particular membrane side was not possible.

#### 2.4.3. SEM Analysis of Adhered Bacteria

SEM micrographs enabled a qualitative, side-dependant analysis of bacterial adhesion and assessment of individual contributions from particular topographical features discussed in chapter 3.2.2. Around 60 images of different magnifications were examined, and ten representative ones are shown in Figure 7. Figure 7a–c suggests that macro-waviness present in corrugated PTFE-B areas is a roughness feature on a too large length-scale for the bacteria to “feel” it: the images show bacteria adhered on peaks as well as in valleys. On the other hand, the clusters of bacteria seem to be concentrated on the threads spread over the waves. In areas without threads, defined by nano-scale roughness (Figure 7d,e), the bacteria seemed to prefer smoother plateaus. Regarding the PTFE-W, the images of surface area containing “warts” (Figure 7f–h) revealed that bacteria clustered around them. In areas without the “warts” (Figure 7i,j), the number of bacteria seemed diminished. The clusters were more frequently observed within the holes of few-micrometre-length-scale size (Figure 7i), but it was not uncommon to see very small clusters or even individual bacteria on a smooth PTFE-W surface (7j). Interestingly, bacteria did not colonize the same length-scale-sized surface cracks, even in the more advanced stages of adhesion and growth (data not shown). It seems that the nano-scale roughness present within the cracks was not favourable for adhesion, which is similar to the conditions shown in Figure 7d,e for the PTFE-B membrane.

## 3. Discussion

The object of this study was to analyze the differences in early stage bacterial adhesion on two commercial d-PTFE membranes used in dental practice for guided bone regeneration. Our results showed that both membranes were chemically equal (pure PTFE), but differed in a degree of crystallinity, crystallite size, crystallite morphology, in a micrometre surface roughness by an order of magnitude (note that the macroscale surface indentations on PTFE-W are three orders of magnitude larger and were therefore excluded from micrometre-scale roughness analysis), as well as in nano-scale roughness. We established that all four studied bacterial species adhered better to the PTFE-W membrane (Figure 6B). The AFM results (Figure 3, Table 1) showed that this membrane had a lower micro-scale surface roughness, which—albeit counterintuitive—is in accordance with previous findings that the roughness positively influences the biofilm formation, yet presents only a minor contribution to initial adhesion [11]. Nonetheless, the literature on this topic is abundant with conflicting results [5], emphasizing the crucial importance of specifying the length-scale of roughness to avoid confusion. Our results indicate that the nanometre-scale roughness is the major contributing factor for early bacterial adhesion: the smoother PTFE-W surface resulted in stronger early adhesion. The explanation of this result can be found in the recent work of Lazzini et al. [68], in which a model for the initial stage of bacterial adhesion on textured surfaces has been developed based on molecular dynamics. The model highlights the antagonistic role of topographical features whose sizes are smaller compared to the individual bacterium, due to the mechanical stress on the bacterial cell wall due to deformation upon interaction with them. The nano-scale corrugations on the PTFE-B surface (Figure 7e), absent from the PTFE-W surface (Figure 7h), qualify as such features.

Nano-scale smoothness of the PTFE-W surface is a consequence of its high degree of crystallinity (78.6% vs. 34.2% for PTFE-B). However, along with smoothness, higher crystallinity also increases the surface stiffness, which is also a known contributor to early adhesion [5]. The majority of studies reported a positive correlation between adhesion and stiffness [69,70]; hence, higher crystallinity of PTFE-W might have a twofold effect in enhancing early adhesion. Our results are in agreement with the study of Trobos et al. [15], where a higher number of bacteria was established for smoother d-PTFE membrane compared to different types of topographically complex e-PTFE materials.

Our results also revealed that one side of each type of membrane contained micrometre-sized topological artefacts: “warts” on PTFE-W and thread-like structures on PTFE-B. Both artifacts are pure PTFE in different forms arising from distinct production processes [28,30,45]. The results indicated that these artifacts present a major contribution to early bacterial adhesion. As described in the Introduction, early stages of bacterial adhesion include transport of bacteria to a substratum surface, and it is reasonable to assume that fluid dynamics, i.e., convection, is one of the mechanisms of transport [5,11], is affected by these relatively large artefacts. Through bacterial mechanosensing, they might enhance the transition from planktonic to surface-associated state [9], and/or provide protection from shear forces to make the change from the weak initial adhesion to permanent attachment occur more easily and more frequently [2]. Their influence might also be viewed as similar to “co-adhesion”, a phenomenon describing the slowing down of suspended particles by those already attached to the surface, thus increasing the probability of adhering [11]. Furthermore, the abovementioned model of Lazzini et al. [68] also showed that “protrusions larger than the bacterial size may offer a larger contact area and at the same time shelter against the hydrodynamic shear flow, eventually promoting adhesion of cells.”

The roughness and the artifacts had the same effect on all studied bacterial species since all of them showed stronger adhesion on PTFE-W. However, the results demonstrated the differences in relative adhesion strength between the species, thus species–specific interaction must be considered. It is well known that the attachment of bacteria to surfaces is modified by various factors, among which surface free energy plays a major role [3,5,13,14,71]. The study of Zhang et al. [14] found that the adhesion of bacteria is inversely proportional to the SFE difference between a bacterium and a surface. Our results are in good agreement with these predictions: the largest SFE difference was established for *A. actinomycetemcomitans* (Figure 5), the species that demonstrated the weakest adhesion on all the examined PTFE surfaces. However, we must emphasize that the strain used in this research is the laboratory strain that forms opaque smooth (OS) colonies. It is known that *A. actinomycetemcomitans* clinical strains form the transparent rough (TR) colony, which converts to OS by multiple cultivations in a laboratory. This conversion results in the absence of the expression of rough colony proteins (RCP), that are closely related to adhesion on abiotic surfaces. Additionally, the diminishing of fimbria also occurs, which additionally weakens the adhesion [72,73]. For the rest of the species, similar SFE values were found, yet the adhesion on all surfaces was the strongest for *S. oralis*, followed by *V. parvula*, *S. mutans* and *A. actinomycetemcomitans*, indicating the influence of additional factors. The strongest adhesion for *S. oralis* is most likely related to its shortest generation time, which is advantageous during “the race for surface” and makes it a successful early colonizer [15,33]. Another contribution might be the electron-acceptor (acidic) character established for *S. oralis* by MATS assay. Bacterial acidic/basic character would not be considered relevant for adhesion to a perfect PTFE surface, since the SFE of a perfect PTFE surface arises exclusively from the Lifshitz–van der Waals component, or in other words, it has no electron-donor/acceptor abilities. However, our SFE measurements showed a weak basic character for one side of each membrane (Appendix A). Indeed, if *A. actinomycetemcomitans* is excluded for the aforementioned reasons, the strength of the adhesion follows the bacteria acidic character (Figure 6A), indicating that acid–base interactions should not be completely excluded as a contributing factor for bacterial adhesion to real PTFE membranes.

The coating of the membranes with artificial saliva (mimicking acquired oral pellicle) did not change the main result established for the uncoated membranes, which is that all studied bacterial species adhered better on the PTFE-W membrane. This result suggests that the major contributors to early bacterial adhesion recognized for uncoated membranes are also valid in an environment resembling the one in the oral cavity. The results also showed that the coating generally diminished the adhesion strength; only *S. mutans* adhered stronger to coated PTFE membranes. Note that even though the effect of the coating was heterogeneous for different bacterial species, the overall ranking of the species remained unchanged. A general reduction in early adhesion strength by the acquired pellicle is a known phenomenon, attributed to multiple factors, among which is the presence of antibacterial components such as lysozyme and peroxidases [3,74]. Since artificial saliva does not contain those components, we believe that the main contribution is the leveling out and masking of the artifacts’ effect. Stronger adhesion of *S. mutans* to coated membranes can be attributed to its known ability to synthesize adhesins that can bind specifically to glycoproteins, especially mucins [75]. Additionally, artificial saliva contained Ca^2+^ ions, which can serve as a bridging agent in the adhesion of *S. mutans* [76].

## 4. Materials and Methods

### 4.1. PTFE Membranes

Two kinds of industrial non-resorbable textured high-density PTFE membranes were used in this study: (1) Cytoplast^TM^ TXT-200 (Osteogenics Biomedical, Lubbock, TX, USA), labeled PTFE-W (white); (2) Permamem^®®^ (Botiss biomaterials, Zossenm, Germany), labeled PTFE-B (blue). The thickness of the membranes was 200 μm and 80 μm, respectively.

### 4.2. Bacterial Strains and Inoculum Preparation

For the in vitro experiments, *Streptococcus mutans* ATCC 25175, *S. oralis* ATCC 6249, *Aggregatibacter actinomycetemcomitans* ATCC 29522 and *Veilonella parvula* ATCC 10790 (Microbiologics, St Cloud, MN, USA) were used. All bacterial strains used were cultured on blood agar (Biolife, Milan, Italy) with the addition of 5% sheep blood (Biognost, Zagreb, Croatia) under anaerobic conditions at 37 °C for 24–48 h. Pure bacterial cultures were suspended in a modified, protein-rich, BHI (Brain Heart Infusion, Becton, Dickinson and Company; Sparks, MD, USA) medium supplemented with 2.5 g/L mucin (Oxoid, Basingstoke, UK), 1.0 g/L yeast extract (Oxoid, Basingstoke, UK), 0.1 g/L cysteine (Sigma-Aldrich, Burlington MA USA), 2.0 g/L sodium bicarbonate (Merck, Darmstad, Germany), 5.0 mg/mL hemin (Sigma-Aldrich, Burlington MA USA), 1.0 mg/mL menadione (Merck, Darmstad, Germany) [77]. Bacteria were cultivated anaerobically until an early stationary growth phase was reached. The log-phase bacteria were further diluted to bacterial suspensions of appropriate concentration (10^7^ or 10^8^ CFU/mL) by measuring the optical density at 600 nm (OD_600_).

### 4.3. Bacterial Adhesion to PTFE Membranes

Sterile, non-coated PTFE membranes and saliva-coated membranes were used to test bacterial adhesion. Membranes were placed in wells of 96-well microtiter plates, part of the membrane was conditioned for 4 h at 30 °C in 50% artificial saliva with composition as described elsewhere [78]. Briefly, artificial saliva included porcine stomach mucins (Sigma-Aldrich, Burlington MA USA) (0.25% *w*/*v*), sodium chloride (Chem-Lab NV, Zedelgem, Belgium) (0.35 *w*/*v*), potassium chloride *p.a*. (Kemika, Zagreb, Croatia) (0.02 *w*/*v*), calcium chloride dihydrate *p.a.* (Kemika, Zagreb, Croatia) (0.02 *w*/*v*), yeast extract (Biolife, Milan, Italia) (0.2 *w*/*v*), lab lemco powder (Oxoid, Basingstoke, UK) (0.1 *w*/*v*), proteose peptone (Biolife, Milan, Italia) (0.5 *w*/*v*) in ddH_2_O (Sigma-Aldrich, Burlington MA USA), and urea *p.a.* (Kemika, Zagreb, Croatia) 0.05% (*v/v*). Artificial saliva was removed and 200 µL of individual bacterial suspension in protein-rich, BHI liquid medium was added in a concentration of 10^7^ CFU/mL. After incubation under anaerobic conditions for 4 h, the planktonic bacteria in the medium were removed and membranes were washed 3× in sterile saline solution. Adhered bacteria were detached by treatment in an ultrasonic bath (Bactosonic, Bandelin, Germany) at 40 kHz for 1 min. In order to quantify the adherent bacteria, ten-fold dilutions were plated on blood agar and CFU/mL was determined.

### 4.4. Attenuated Total Reflection-Fourier Transform Infrared (ATR-FTIR) Spectroscopy

ATR-FTIR spectroscopy was used to investigate the chemical composition of the two membranes. Spectra were recorded using Spectrum Two ATR-FTIR spectrometer (PerkinElmer Inc., Waltham, MA, USA) equipped with a LiTaO_3_ detector type. The spectra were corrected for background and recorded (8 scans) at 3 selected locations on each sample in the absorbance mode, in a wavelength region from 600 cm^–1^ to 4000 cm^–1^ at a resolution of 0.5 cm^–1^. The relevant absorption bands were interpreted using Spectrum V.10.5.3 software (PerkinElmer Inc.).

### 4.5. Differential Scanning Calorimetry (DSC)

Thermal analysis was performed primarily to examine the differences in membranes’ crystalline phase. DSC measurements were carried out using a Mettler-Toledo DSC822e calorimeter calibrated with an indium standard under a nitrogen gas atmosphere. The membrane samples of about 5 mg were analyzed either as received (dry PTFE) or after soaking at room temperature for 48 h in 50% artificial saliva (wet PTFE). The heating and cooling rates were 10 °C/min and 30 °C/min, respectively. The degree of crystallinity was obtained as a mean value of three measurements, from the melting peak area of the first heating step using 82.0 J/g (4.10 kJ/mol) as melting enthalpy of the 100% crystalline sample [79].

### 4.6. X-ray Diffraction (XRD)

XRD diffractograms were obtained in order to analyze the differences in crystalline phase features such as average crystallite size and *d*-spacing. A Philips vertical goniometer (type X’Pert) equipped with a Cu tube was used with the following experimental conditions: 45 kV, 40 mA, PW 3018/00 PIXcel detector, primary beam divergence 1/4°, continuous scan (step 0.02). The interpretation and the quantitative analysis of diffractograms were obtained using HIGH SCORE PLUS (2016) calculation. The average crystallite size was calculated using the Scherrer equation *L* = *K λ/*(*b* cos *θ*), where *b* is the FWHM of the peak (in rad), *λ* is the wavelength of the X-rays used (1.5418 Ǻ for Cu Kα radiation), *θ* is the angle which is calculated by taking 1/2 of 2*θ* value. *K* is a constant of proportionality (the Scherrer constant) of the order of unity. Its exact value depends on various factors such as the crystallite shape and the crystallite-size distribution. The value of *K* is 0.9 for FWHM of spherical crystals with cubic symmetry [40].

### 4.7. Scanning Electron Microscopy (SEM) Analysis of Adherent Bacteria on PTFE Membranes

The analyses of the adherent bacteria on d-PTFE membranes after 4 h incubation were performed using Jeol JSM-7800F (JEOL Ltd., Tokyo, Japan) field emission scanning electron microscope (FE-SEM). Prior to FE-SEM observations, the membranes were incubated for 4 h in bacterial suspensions, washed in sterile PBS and air-dried in a high-flow sterile chamber. After fixation with 4% glutaraldehyde and 0.5% paraformaldehyde (Sigma-Aldrich, Burlington, MA, USA) prepared at 4 °C in 0.1 M PBS (Sigma-Aldrich, Burlington, MA, USA) (pH 7.2) and subsequent dehydration by immersion in a series of increasing concentrations of ethanol (50, 70, 80, 90 and 100%, Sigma-Aldrich, Burlington, MA, USA), the membranes were mounted on the carbon tape attached to the sample holder. To increase surface conductivity and stability of the membranes under electron beam exposure in high vacuum conditions, the samples were sputter-coated with the thin 15 nm layer of gold-palladium using precision etching and coating system PECS II (Gatan Inc. Pleasanton, CA, USA).

### 4.8. Atomic Force Microscopy (AFM)

The surface topology scans of the membranes were obtained using the atomic force microscope Bruker Dimension Icon (Bruker, Berlin, Germany) in tapping mode. Tapping mode was employed in order to obtain a high-resolution scans of surface details, using a Bruker SNL-10 type D (low stiffness) silicon nitride cantilever with a silicon tip of 2 nm tip radius. The cantilever’s dynamic properties, such as natural frequency and normal stiffness, were obtained using the thermal tune method where the thermal noise spectrum of the cantilever was measured and fitted to a Lorentzian harmonic oscillator model in the air, which corresponded to 22 kHz. The obtained calibration allows the usage of minimal contact force for the measurement, which has minimal impact on the surface, and thus the average used force was 1.5 pN. ±5%. The scans were made with scan sizes of 2 µm^2^ and 5 µm^2^ with 512 scan lines each, with 512 data points acquired per line. The obtained data were processed in order to obtain the values of surface roughness parameters after tilt and bow corrections using the proprietary Bruker Nanoscope Analysis software (Bruker, Berlin, Germany). The roughness is given as root-mean-square (RMS) roughness, defined as the standard deviation of the elevation within the given area.

### 4.9. Digital Microscopy

The surfaces of the membranes were examined microscopically using a digital microscope (DM) DSX 1000 (Olympus, Tokyo, Japan) at 20× magnification. Analyses were performed at 3 different locations (1 mm × 1 mm) for each membrane. The DM provided the 2-dimensional and 3-dimensional images of the analyzed areas.

### 4.10. Microbial Adhesion to Solvents (MATS)

Bacteria surface hydrophobicity and acid–base properties were measured using the Microbial Adhesion to Solvents (MATS) assay technique. The affinity of bacterial cells to three different solvents, namely chloroform *p.a* (Kemika, Zagreb, Croatia), diethyl ether *p.a.* (T.T.T, Sveta Nedjelja, Croatia) and n-hexane, 98% *p.a.* (GRAM-MOL, Croatia), was used. An 18–20 h bacterial culture was prepared in a protein-rich BHI medium. The bacterial suspension was centrifuged for 10 min at 7000× *g* and 4 °C, washed 2×, and resuspended in sterile saline to a concentration of 10^8^ CFU/mL. 0.2 mL of the test solvent was added to 1.2 mL of the bacterial suspension and shaken vigorously for 90 s. The emulsion was allowed to stand for 15 min, the time required to achieve phase separation, in order to separate the aqueous phase sample and measure the optical density at 600 nm (OD_600_). All measurements were performed in triplicate. Adhesion percentage, i.e., the affinity of microorganisms to the solvents, was calculated as follows: adherence (%) = [1 − *A*/*A*_0_] × 100. *A*_0_ and *A* are the optical densities of the bacterial suspension before and after the mixing with the solvent, respectively [80].

### 4.11. Surface Free Energy (SFE) Determination

#### 4.11.1. PTFE Membranes

SFE of the membranes was determined by contact angle (CA) measurement technique using the DataPhysics OCA Instruments GmbH (Filderstadt, Germany), equipped with an automated dosing system. The sessile drop method was used. The CA analysis was performed with a software SCA 20, Version 2.01 (DataPhysics Instruments GmbH, Filderstadt, Germany) using three models: (1) Wu’s model [57]; (2) Owens–Wendt’s model [56]; (3) the van Oss–Chaudhury–Good model, i.e., acid–base model [10]. The former two models give the dispersive ($\gamma^{d}$) and the polar ($\gamma^{p}$) SFE components. In the acid–base model, the dispersive component is referred to as the Lifshitz–van der Waals component ($\gamma^{\mathrm{LW}}$), while the polar contribution is subdivided into Lewis acid ($\gamma^{+}$) and Lewis base ($\gamma^{-}$) components. The detailed descriptions of the models can be found in references [1,11,46].

Three probe liquids were used, namely, water, formamide (>99.5%, Fluka, Buchs, Switzerland) and diiodomethane (99+ %, Acros Organics, Geel, Belgium). Dispersive and polar components of probe liquids SFEs are given in Table 3. These values were used for membranes SFE calculation using Wu, Owens–Wendt, and acid–base models. All measurements were carried out at 25 °C, with a drop volume of 2 μL. The mean CA value for each probe liquid was obtained by measuring 10–15 drops. The deviation from the mean was ± 3°.

#### 4.11.2. Bacterial Cells SFE

Bacterial SFE was determined by a spectrophotometric method proposed by Zhang et al. [14,71]. A series of predefined ratios of pure ethanol *p.a*. (Kemika, Zagreb, Croatia) and ultrapure water with surface tensions ranging from 22 to 72 mJ/m^2^ was prepared. The bacterial suspension was centrifuged, washed three times in PBS (Sigma-Aldrich, Burlington, MA, USA), resuspended to a concentration of 10^9^ CFU/mL, and then vortexed for 1 min to a homogeneous mixture. To 0.5 mL of each medium in series, 10 µL of bacterial suspension was added, stirred vigorously, and allowed to stand for 20 min. This was followed by centrifugation for 6 min at 100× *g* to separate the supernatant from the sediment. 200 μL of the supernatant was transferred to a 96-well microtiter plate and optical density at 595 nm (OD_595_) was measured with a microtiter plate reader (BioTek EL808, Santa Clara, CA, USA). OD_595_ was plotted against the surface tension of the liquid medium. All measurements were performed in triplicate [14].

### 4.12. Statistical Analysis

Data were expressed as means and standard deviation. The obtained results were evaluated using the Mann–Whitney U test for comparison of two groups or Kruskal–Wallis ANOVA by Ranks for comparing multiple groups. Results were considered statistically significant at *p* < 0.05. The statistical analysis was performed in StatisticaTM Software (version 14.0.0.15, TIBCO Software Inc, Palo Alto, CA, USA).

## 5. Conclusions

Two types of commercial d-PTFE membranes had the same chemical composition, but their surface microstructures greatly differed. The most pronounced differences were observed in the degree of crystallinity: as opposed to PTFE-B, PTFE-W was highly crystalline, with a large average crystallite size and two crystallite morphologies (dendrites and “warts”). PTFE-W was also smoother on the nanometre scale, which was attributed to the high degree of crystallinity. Both membranes contained topographical artifacts such as spherical crystallites and thread-like structures. All four bacterial species studied in this work demonstrated stronger early adhesion on the PTFE-W membrane, both with and without the artificial saliva. The strongest early adhesion was established for *S. oralis*, followed by *V. parvula*, *S. mutans* and *A. actinomycetemcomitans*.

Our results provide the contribution to understanding the adhesion of early oral microbial colonizers that can assist clinicians in choosing the membrane for GBR procedure. Finally, our results could help in improving antimicrobial properties of PTFE membranes through tailoring the polymer microstructural properties.

## Figures and Tables

**Figure 1 ijms-23-02983-f001:**
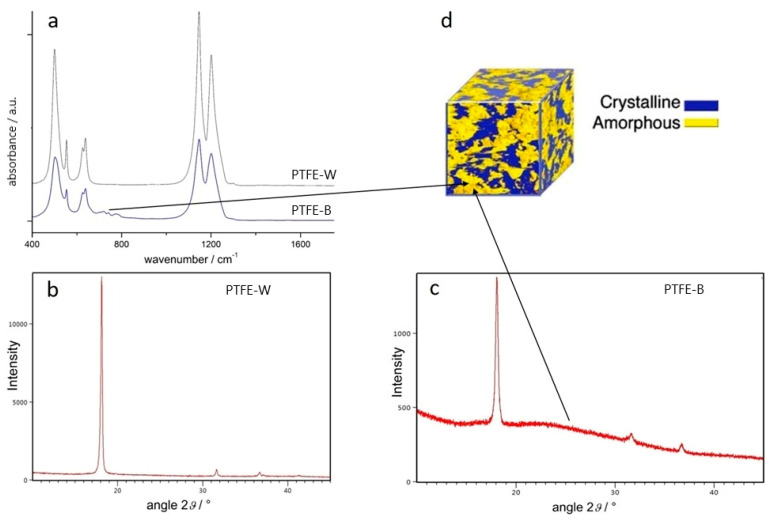
Fourier transform infrared (FTIR) spectra (**a**) and X-ray diffractograms (**b**,**c**) of polytetrafluoroethylene (PTFE) membranes; (**d**) illustration of the semi-crystalline nature of PTFE (image adapted with permission from Brown et al. [38]). PTFE-W: polytetrafluoroethylene-white; PTFE-B: polytetrafluoroethylene-blue.

**Figure 2 ijms-23-02983-f002:**
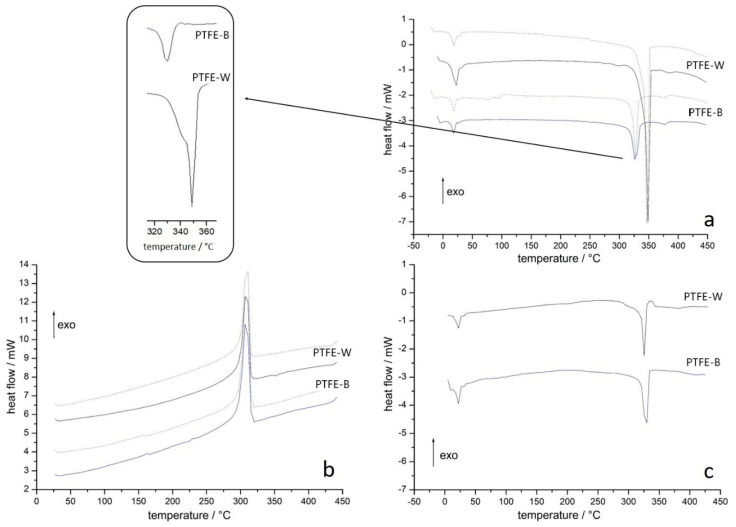
Differential scanning calorimetry (DSC) thermograms for (**a**) 1st heating run, (**b**) cooling run and (**c**) 2nd heating run of as-received membranes (solid lines) and after soaking in artificial saliva for 48 h (dotted lines). PTFE-W: polytetrafluoroethylene-white; PTFE-B: polytetrafluoroethylene-blue.

**Figure 3 ijms-23-02983-f003:**
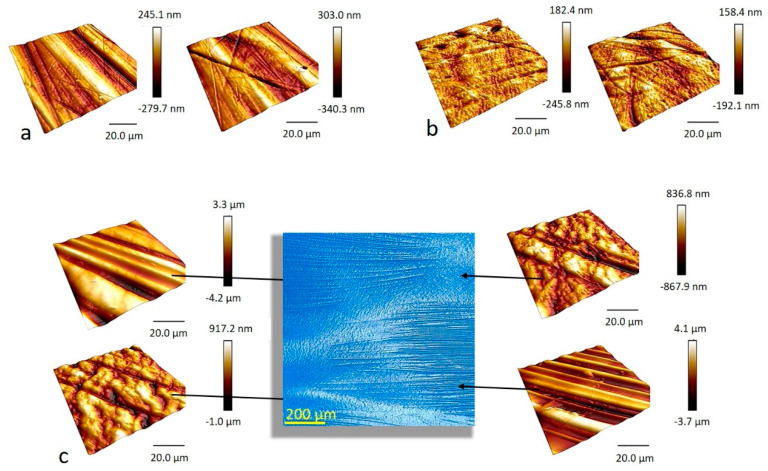
Microscale images of PTFE membranes. (**a**) PTFE-W (polytetrafluoroethylene-white) side 1 and (**b**) PTFE-W side 2; (**c**) PTFE-B (polytetrafluoroethylene-blue), showing the areas with and without macro-waviness, present on both sides of PTFE-B membrane (presented by a digital microscope image).

**Figure 4 ijms-23-02983-f004:**
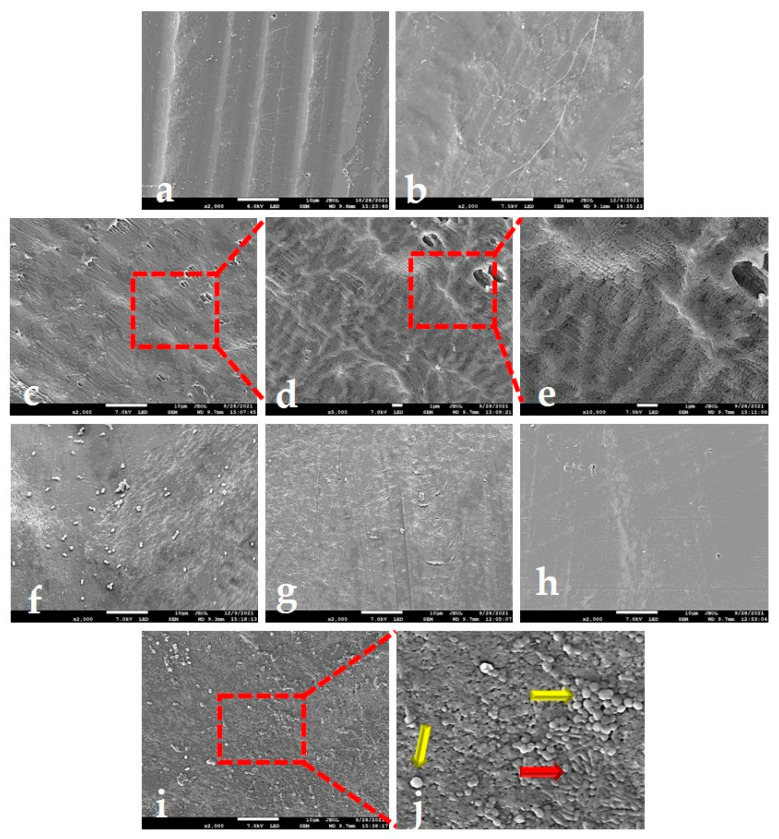
Representative scanning electron microscopy (SEM) micrographs of PTFE membranes: (**a**,**b**) PTFE-B (polytetrafluoroethylene-blue) side 1, showing areas with and without macro-waviness, and fine fibers distributed over them; (**c**–**e**) PTFE-B side 2, shown with different magnification; crystallites in a form of dendrites are visible as a dominant nano-structural feature; fine fibers are not visible on this side. (**f**–**h**) PTFE-W (polytetrafluoroethylene-white) surfaces, revealing heterogeneously distributed spherical crystallites or “warts” on an otherwise smooth surface; (**i**,**j**) PTFE-W micrographs showing the coexistence of “warts” (yellow arrow) and dendrites (red arrow).

**Figure 5 ijms-23-02983-f005:**
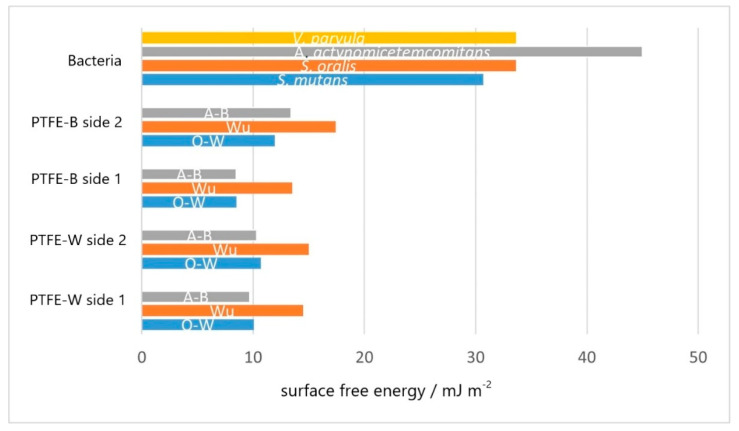
Total surface free energy of bacteria and each side of the PTFE membranes calculated using the Owens–Wendt (O–W), Wu, and acid–base (A–B) model. PTFE-B (polytetrafluoroethylene-blue); PTFE-W (polytetrafluoroethylene-white).

**Figure 6 ijms-23-02983-f006:**
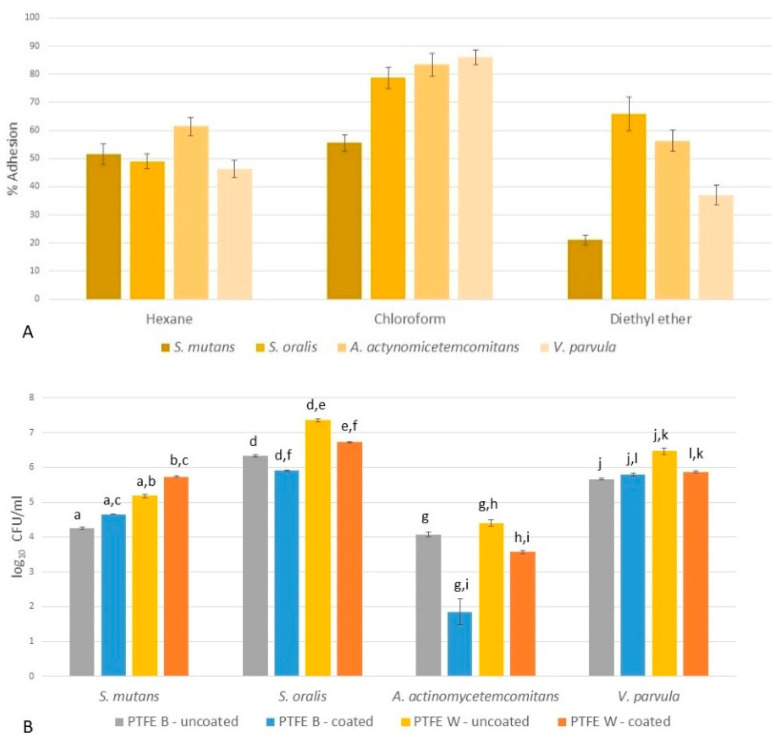
(**A**) Bacterial adhesion to hexane, chloroform and diethyl ether, demonstrating their hydrophobic, electron-donor (Lewis base) and electron-acceptor (Lewis acid) character. (**B**) Bacterial colonization after 4 h on uncoated PTFE-W (polytetrafluoroethylene-white) and PTFE-B (polytetrafluoroethylene-blue) surface, and on a surface coated with artificial saliva. (a–l) Comparison within each bacterium species and different surfaces showed that the differences were statistically significant (*p* > 0.05).

**Figure 7 ijms-23-02983-f007:**
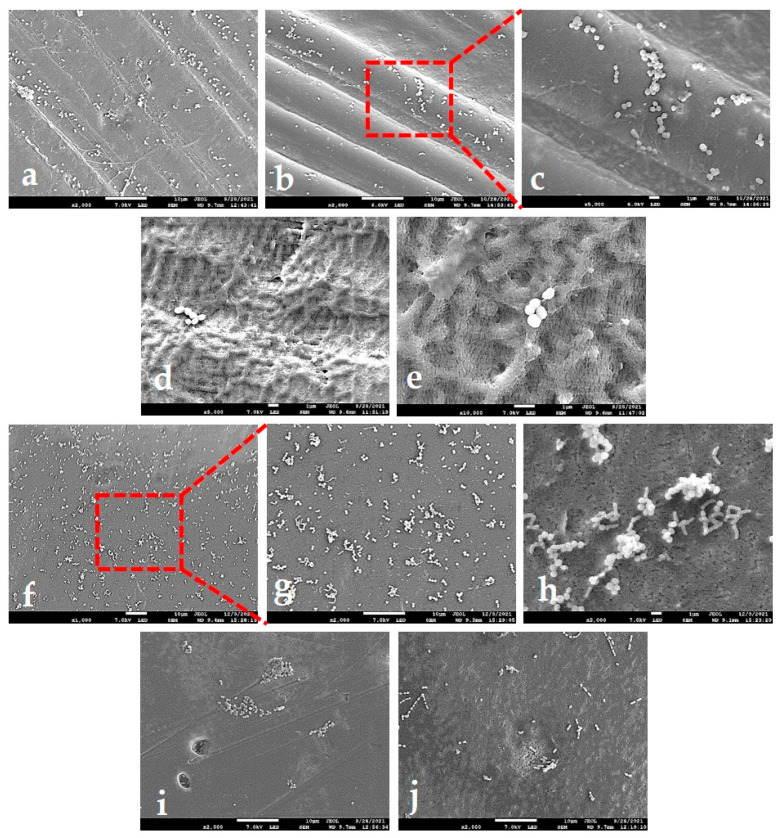
Representative SEM micrographs of bacteria adhering to PTFE membranes. (**a**–**c**) different magnifications of the corrugated area of PTFE-B (polytetrafluoroethylene-blue) membrane, containing thread-like structures; (**d**,**e**) bacteria adhering to PTFE-B in an area without thread-like structures; (**f**–**h**) different magnifications of “warts”-rich area of PTFE-W (polytetrafluoroethylene-white) membrane; (**i**,**j**) bacteria adhering on areas without “warts”.

**Table 1 ijms-23-02983-t001:** Root-mean-squared (RMS) values of microscale surface roughness for PTFE membranes.

	RMS/nm	Average RMS/nm
PTFE-W side1a	90.3	83.0
PTFE-W side1b	75.6	
PTFE-W side2a	50.8	49.2
PTFE-W side2b	47.5	
PTFE-B side1a	281	693.5
PTFE-B side1b	1106	
PTFE-B side2a	252	735.0
PTFE-B side2b	1218	

**Table 2 ijms-23-02983-t002:** Contact angle values of the probe liquids on PTFE membranes used for SFE calculations.

	PTFE-W		PTFE-B	
	SIDE 1	SIDE 2	SIDE 1	SIDE 2
water	127.1 ± 2.6	120.9 ± 2.7	127.8 ± 3.7	122.5 ± 4.9
formamide	106.5 ± 2.8	109.3 ± 2.6	113.4 ± 3.8	110.1 ± 2.2
diiodomethane	95.7 ± 5.0	95.8 ± 1.5	99.3 ± 2.8	88.3 ± 4.1

**Table 3 ijms-23-02983-t003:** Surface free energy components (mJ/m^2^) of probe liquids used for calculation using (a) Wu and Owens–Wendt models and (b) acid–base model.

	(a) Dispersive $\gamma_{l}^{d}$	Polar $\gamma_{l}^{p}$	(b) Lifshitz–van der Waals $\gamma_{l}^{LW}$	Acid $\gamma_{l}^{+}$	Base $\gamma_{l}^{-}$	Total SFE $\gamma_{l}$
water	21.8	51.0	21.8	25.5	25.5	72.8
formamide	39.0	19.0	39.0	2.28	39.6	58.0
diiodomethane	50.8	0.00	50.8	0.00	0.00	50.8

## Data Availability

Not applicable.

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
