# Peer review of "Adhesion of Oral Bacteria to Commercial d-PTFE Membranes: Polymer Microstructure Makes a Difference"

_ijms, 2022, doi:10.3390/ijms23062983_

Round 1
Reviewer 1 Report
Reviewed manuscript number 1622587 entitled "Adhesion of oral bacteria to commercial d-PTFE membranes: polymer microstructure makes a difference" treats about the early stages of different choosen bacterial adhesion on two commercial dense polytetrafluoroethylene (d-PTFE) membranes. The studied phenomena is very atual according to the practical application of the PTFE. Authors studied and connected the microstructural features with the adhesion strengths of bacteria. The results indicated that the degree of crystallinity is the principal feature promoting the adhesion strength, through lower nanoscale roughness and possibly higher surface stiffness. The presented studies results are well organized and presented, and could be very interesting for the readers. In my opinion the manuscript could be suitable for publication in IJMS following the changes suggested below:
- Especially the method for the contact angle measurements to the SFE determination is disscusive. The roughness of PTFE membranes is quite high. It could influence on the contact angle values in higher degree. In addition the bacterials surface tension was determinad in different way. The contact angle chanches resulting from the PTFE roughness could be neglected and the SFE of PTFE and bacteria values could be comparable only if it will be detrmined in the same way. It should be more explained. It should be also remembered that even for the model liquids used for the SFE determination the surfece tension data should be determined in the same way. Please see ref. Applied Suirface Sci. 2017,405, Pages 88-101.
Author Response
Response to Reviewer #1 of the manuscript ID: ijms-1622587 entitled: “Adhesion of oral bacteria to commercial d-PTFE membranes: polymer microstructure makes a difference″, by Gabrijela Begić, Mirna Petković Didović, Sanja Lučić Blagojević, Ivana Jelovica Badovinac, Jure Žigon, Marko Perčić, Olga Cvijanovic Peloza and Ivana Gobin″
General comments:
Reviewed manuscript number 1622587 entitled "Adhesion of oral bacteria to commercial d-PTFE membranes: polymer microstructure makes a difference” treats about the early stages of different chosen bacterial adhesion on two commercial dense polytetrafluoroethylene (d-PTFE) membranes. The studied phenomena is very actual according to the practical application of the PTFE. Authors studied and connected the microstructural features with the adhesion strengths of bacteria. The results indicated that the degree of crystallinity is the principal feature promoting the adhesion strength, through lower nanoscale roughness and possibly higher surface stiffness. The presented studies results are well organized and presented, and could be very interesting for the readers. In my opinion the manuscript could be suitable for publication in IJMS following the changes suggested below:
Response:
The Authors would like to thank Reviewer #1 for her/his valuable comments on our manuscript. We have responded to the comments and revised the paper accordingly.
Comment 1:
Especially the method for the contact angle measurements to the SFE determination is disscusive. The roughness of PTFE membranes is quite high. It could influence on the contact angle values in higher degree. In addition, the bacterials surface tension was determinad in different way. The contact angle chanches resulting from the PTFE roughness could be neglected and the SFE of PTFE and bacteria values could be comparable only if it will be detrmined in the same way. It should be more explained. It should be also remembered that even for the model liquids used for the SFE determination the surfece tension data should be determined in the same way. Please see ref. Applied Suirface Sci. 2017,405, Pages 88-101.
Response:
Thank you for your observation. The description of challenges related to SFE measurements of solid surfaces is discussed in the first section of Chapter 2.3 and now further clarified according to your suggestion. Bacterial SFE was determined using the method proposed by Zhang et al. [14], since their study showed that the difference between bacterial SFE (determined spectrophotometrically) and substratum SFE (determined by contact angle measurements) is the crucial parameter that dictates the adhesion strength. This was exactly what we were interested in in our study, so we adopted the proposed approach in its full.

Reviewer 2 Report
The article is well designed and provides an answer to the question of bacterial adhesion to two types of membranes.
I am not competent to comment on the method just mentioned. I think clinicians can find the answer to which type of membrane to use.
Author Response
Response to Reviewer #2 of the manuscript ID: ijms-1622587 entitled: “Adhesion of oral bacteria to commercial d-PTFE membranes: polymer microstructure makes a difference″, by Gabrijela Begić, Mirna Petković Didović, Sanja Lučić Blagojević, Ivana Jelovica Badovinac, Jure Žigon, Marko Perčić, Olga Cvijanovic Peloza and Ivana Gobin″
The article is well designed and provides an answer to the question of bacterial adhesion to two types of membranes.
I am not competent to comment on the method just mentioned. I think clinicians can find the answer to which type of membrane to use
Response:
The Authors would like to thank Reviewer #2 for her/his generous comment on our manuscript.

Reviewer 3 Report
Comment 1: Qualitative informations are missing in abstract. Abstract should be concise and the authors need to improve with more specific short results.
Comment 2: In abstract, infrared (IR) should be revised as Fourier transform infrared (FTIR).
Comment 3: The figures in the manuscript were poor, the author should improve the quality and solution of these figures.
Comment 4: Materials and Methods part should be mentioned before results and discussion
Comment 5: Compare your results with literature ones.
Comment 6: The purity of used products should be added.
Comment 7: The introduction section should be modified though citing recent references (2020, 2021 and 2022) related studies and indicating the novelty of the study compared to the carried works. Also, the following references should be included in the introduction part.
- Composite Structures 262 (2021) 113640 https://doi.org/10.1016/j.compstruct.2021.113640
- Heliyon 6 (2020) e04187 (https://doi.org/10.1016/j.heliyon.2020.e04187).
- SN Applied Sciences 1 (2019) 1-9 (https://doi.org/10.1007/s42452-019-0911-8).
- Journal of King Saud University Science 32 (2020) 235-244 (https://doi.org/10.1016/j.jksus.2018.04.030).
- Polymer Bulletin 76 (9) (2019) 4859-4878 (https://link.springer.com/article/10.1007/s00289-018-2639-9).
Comment 8: Conclusion is too long. Conclusion should be revised and improved.
Author Response
Response to Reviewer #3 of the manuscript ID: ijms-1622587 entitled: “Adhesion of oral bacteria to commercial d-PTFE membranes: polymer microstructure makes a difference″, by Gabrijela Begić, Mirna Petković Didović, Sanja Lučić Blagojević, Ivana Jelovica Badovinac, Jure Žigon, Marko Perčić, Olga Cvijanovic Peloza and Ivana Gobin″
Comment 1: Qualitative informations are missing in abstract. Abstract should be concise and the authors need to improve with more specific short results.
Response:
The Reviewer’s suggestions have been accepted. The abstract has been refined in such a way that we have highlighted the most important results in our manuscript.
Comment 2: In abstract, infrared (IR) should be revised as Fourier transform infrared (FTIR).
Response:
The Reviewer’s suggestions have been accepted. " Infrared (IR) " was replaced with " Fourier transform infrared (FTIR)".
Comment 3: The figures in the manuscript were poor, the author should improve the quality and solution of these figures.
Response:
Thank you for the suggestion. The image quality is adjusted to journal requirements (a figure resolution of 300 dpi or higher). We have improved, according to your suggestion, the size of the images and thus emphasized the result we show on them.
Comment 4: Materials and Methods part should be mentioned before results and discussion
Response:
The article was prepared using the template of the journal IJMS, in which the introduction is followed by results, discussion and materials and methods.
Comment 5: Compare your results with literature ones.
Response:
Thank you for the suggestion, the results are further linked to the literature and the above change is marked in the text.
Comment 6: The purity of used products should be added.
Response:
Apologies for the omission. This has been corrected according to the suggestion.
Comment 7: The introduction section should be modified though citing recent references (2020, 2021 and 2022) related studies and indicating the novelty of the study compared to the carried works. Also, the following references should be included in the introduction part.
- Composite Structures 262 (2021) 113640 https://doi.org/10.1016/j.compstruct.2021.113640
- Heliyon 6 (2020) e04187 (https://doi.org/10.1016/j.heliyon.2020.e04187).
- SN Applied Sciences 1 (2019) 1-9 (https://doi.org/10.1007/s42452-019-0911-8).
- Journal of King Saud University Science 32 (2020) 235-244 (https://doi.org/10.1016/j.jksus.2018.04.030).
- Polymer Bulletin 76 (9) (2019) 4859-4878 (https://link.springer.com/article/10.1007/s00289-018-2639-9).
Response:
Thanks for the suggestion. The introduction section has been modified to the extent that it fits the theme of manuscript.
Comment 8: Conclusion is too long. Conclusion should be revised and improved.
Response:
The Reviewer’s suggestions have been accepted. The conclusion section has been additionally revised and improved.
